# Step-by-Step Development and Implementation of FS-MPC for a FPGA-Based PMSM Drive System

Ipsita Mishra [1,*], Ravi Nath Tripathi [2], Vijay Kumar Singh [3] and Tsuyoshi Hanamoto [1]

1   Department of Life Science and System Engineering, Graduate School of Life Science and Systems Engineering, Kyushu Institute of Technology, Kitakyushu, Fukuoka 808-0196, Japan; hanamoto@life.kyutech.ac.jp

2   Nagamori Actuator Research Center, Kyoto University of Advanced Science, Kyoto 6168577, Japan; tripathi.ravi@kuas.ac.jp

3   Department of Electrical Engineering, National Institute of Technology, Ashok Rajpath, Patna 800005, India; vijay.ee@nitp.ac.in

*   Correspondence: mishra.ipsita530@mail.kyutech.jp or ipsitamishra713@gmail.com; Tel.: +81-80-2137-1661

**Abstract:** In this paper, finite-set model-predictive control (FS-MPC) is inducted for a motor drive system. The dynamic response and multiple constraint handling nature of FS-MPC are the major factors that stand out among the controller family. However, for real-time implementation, the computational burden of FS-MPC is a primary concern. Due to the parallel processing nature and discrete nature of the hardware platform, the field-programmable gate array (FPGA) can be an alternative solution for the real-time implementation of the controller algorithm. The FPGA is capable of handling the computational requirements for FS-MPC implementation; however, the system development involves multiple steps that lead to a time-consuming debugging process. Moreover, specific hardware coding skill makes it more complex, corresponding to an increase in system complexity, which leads to a tedious task for the system development. This paper presents a FPGA-based implementation of the predictive current control of a permanent magnet synchronous motor (PMSM). FS-MPC of the PMSM drive system is designed and implemented using the digital model integration approach provided by the Xilinx system generator (XSG) and VIVADO platform. The step change in the load disturbance as well as the reference speed is considered for the analysis of the controller for the motor drive system. Moreover, the steady state error and harmonic distortion in the motor current is considered for an in-depth analysis of the system performance corresponding to different sampling frequencies.

**Keywords:** finite-set model-predictive control; field-programmable gate array; permanent magnet synchronous motor; Xilinx system generator

## 1. Introduction

Permanent magnet synchronous motors (PMSMs) have been used in high-performance AC motor drives in industrial as well as domestic applications due to the advantages of high efficiency, a high torque-to-weight ratio and a wide range of speed operations [1–3]. The motor drive system needs to achieve the desired speed and a smooth transition corresponding to the change in the motor load condition, depending on its applications. With the progress of power electronics and microprocessor techniques, different control concepts have been proposed to achieve the desired PMSM response.

Conventional control techniques such as field-oriented control (FOC) and direct torque control (DTC) are widely used in industrial applications for PMSM control [4,5]. However, in the case of conventional control, the system complexity increases with the inclusion of the control parameters. DTC has good dynamic performance; however, it has poor steady-state performance. FOC has good steady-state performance compared to that of DTC. Due to the absence of the hysteresis band, the digital implementation of FOC is easier. However, FOC utilizes a cascaded PI controller for its implementation, and there are at

least four parameters it needs to tune separately. With a change in working conditions, the drive performance may worsen [6]. The inner current control loop has a crucial role in motor drive systems to generate the reference voltage, which ultimately generates the switching signal. For small- and medium-power motor drive systems, the current-loop bandwidth is restricted due to a limited switching frequency, which directly impacts the system dynamic performance [7]. Moreover, the motor current is regulated by the terminal voltage applied by an inverter. Above the medium-speed range, if an abrupt change in the reference is applied, the reference voltage generated from the current control loop may undergo saturation, which, finally, leads to degradation in the current regulation [8] and a worsened motor performance. To achieve a fast dynamic response with better current regulation, a better controller is always desirable.

Model predictive control (MPC) is an advanced control strategy. It has become an attractive solution for PMSM control and can improve the dynamic response by predicting the next step current [9]. The MPC possesses attractive features such as direct use of the system model and a simultaneous multiple constraint handling nature to deal with multiple control parameters [10–12]. Considering these appealing characteristics, MPC is gaining attention among researchers and also has been applied in a wide variety of drive applications. The MPC implementation methodology utilizes the mathematical model of the plant to predict its future behavior, and an optimization function is used for the selection of the switching signal. The two main categories of MPC are continuous-control-set MPC (CS-MPC) and finite-control-set MPC (FS-MPC) [13]. CS-MPC generates switching signals to the power converter by calculating a continuous-controlled variable through a modulator with a constant switching frequency [14]. On the other hand, FS-MPC considers the finite set of the possible switching states of the power converter for the generation of switching signals for the power devices [15,16]. Compared with CS-MPC, the major advantage of FS-MPC is that it can directly generate the switching signal by optimizing the control parameter without any modulation stage. Finite-set-predictive torque control (FS-PTC) with discrete space vector modulation (DSVM) for a PMSM drive is presented in [17]. Moreover, a new preselection strategy is proposed to reduce the computational numerations from 37 to 6 voltage vectors, reduce the torque and flux ripples and ultimately achieve robust characteristics against parameter variations. In [18], a total harmonic distortion (THD)-oriented FCS-MPC controller is designed for single-phase inverters. To achieve this goal, a modified cost function is constructed using a linear combination, with weight factors of the current fundamental tracking error and an instantaneous THD value. A FS-MPC strategy with distribution in harmonic spectra similar to a PWM controller is proposed in [19]. This strategy reduces the number of commutations and also fixes the harmonic spectra. However, the cost function optimization problem is computed by predicting all the possible switching states of the power converter in every sampling period. With an increase in the complexity of the converter system, the system may undergo delays, which ultimately degrades the system performance [20,21].

The field-programmable gate array (FPGA) has the advantage of parallel processing, which shortens the computational time, ultimately to leading a decrease in the control delay and better system performance [22,23]. Moreover, the FPGA is considered the better option for controller designing and prototyping due to its fast computation ability, embedded processor and shorter design cycle [24,25]. In [26], an implementation methodology using the hardware description language (HDL) coder from MathWorks is presented with an automated workflow for implementing long-horizon FCS-MPC for a PMSM drive system. At present, a model-based design (MBD) for the FPGA implementation of complex control systems is attractive because of its time-saving capacity and great flexibility in simulation and debugging [27–29].

This paper presents the design and development of FS-MPC for a PMSM drive system that can be utilized in real-time FPGA implementations. The FS-MPC algorithm is developed for a three-phase, two-level voltage source inverter (VSI)-fed PMSM drive system using an optimization function in terms of current. The step-by-step design and devel-

opment of the controller for the motor drive system are explained. The motor dynamics response is analyzed corresponding to different sampling frequencies. To analyze the dynamic behavior of the motor, a step change in the reference speed and load disturbance is introduced. Moreover, for the steady-state analysis, the steady-state error and harmonics in the motor current are considered corresponding to different sampling frequencies. The system performance is validated by comparing it with a linear PI controller. Furthermore, a model-based approach is adopted for the system implementation for the modeling of FS-MPC. For the real-time implementation of the FPGA, the controller is developed in the Xilinx system generator (XSG) environment integrated with MATLAB/Simulink, which automatically generates the HDL code.

The sections of this paper are organized as follows: Section 2 describes the algorithm of FS-MPC for the PMSM drive system considering the discrete-time mathematical model of the motor and three-phase VSI. In Section 3, the model-based design and development of the controller in the XSG environment are explained. A FPGA-based system implementation is explained in Section 4. The real-time implementation of the controller and results, along with a detailed discussion, are presented in Section 5. Finally, Section 6 concludes the paper.

## 2. PMSM Drive

### 2.1. PMSM Mathematical Model

The power circuit with a three-phase inverter-fed PMSM motor is shown in Figure 1. The PMSM stator dynamic equations are described as:

$$v_s = R_s i_s + \frac{d\psi_s}{dt} \tag{1}$$

where $v_s$ and $i_s$ are the stator voltage and current, respectively, and $R_s$ is the stator resistance. $\psi_s$ is the stator flux produced by the stator current and permanent magnet rotor. The equation describing $\psi_s$ is given by:

$$\psi_s = L_s i_s + \psi_m \tag{2}$$

where $L_s$ is the stator self-inductance, and $\psi_m$ is the flux produced by the permanent magnet rotor. The stator equation on the $dq$ coordinate can be given as:

$$v_d = R_s i_d + L_s \frac{di_d}{dt} - \omega_r L_s i_q \tag{3}$$

$$v_q = R_s i_q + L_s \frac{di_q}{dt} + \omega_r L_s i_d + \psi_m \omega_r \tag{4}$$

where, $\omega_r = d\theta/dt$ is the mechanical rotor angular speed.

The electromagnetic torque produced by the motor, which depends on the stator quadrature component current and magnitude of the flux, is expressed below.

$$T_e = \frac{3}{2} p_p \psi_m i_q \tag{5}$$

where $p_p$ is the number of pole pairs in the motor. The mechanical dynamics of the motor can be given as:

$$\frac{d\omega_r}{dt} = \frac{1}{J}(T_e - T_l) - \frac{B}{J}\omega_r \tag{6}$$

where, $J$ is the motor inertia, $B$ is the friction coefficient and $T_l$ is the load torque. The rotor electrical angular speed $\omega_e$ is given by:

$$\omega_e = p_p \omega_r \tag{7}$$

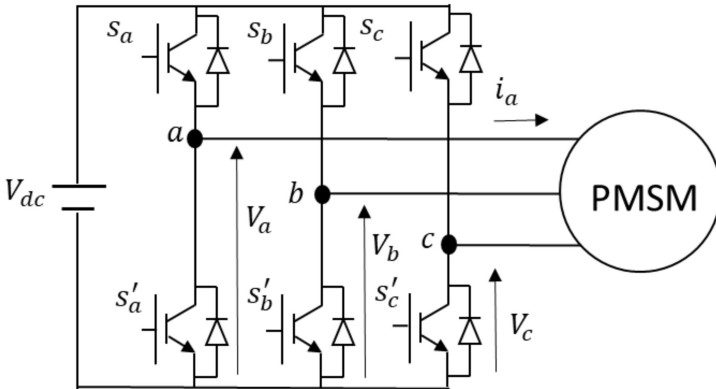

**Figure 1.** Three-phase voltage source inverter (VSI)-fed permanent magnet synchronous motor (PMSM).

## 2.2. FS-MPC for the PMSM Drive

FS-MPC utilizes the discrete-time model of the system and the possible switching states of the three-phase voltage source inverter for the execution of the control algorithm. The total number of switching states corresponding to the three-phase VSI is eight. Corresponding to these eight switching states, eight voltage vectors can be obtained, as given in Table 1. For every possible state of the power devices of the converter, the controller predicts the future behavior of the system variables for the next sampling time. The controller evaluates the values of the cost functions based on these predicted system variables. Finally, corresponding to the minimum cost function, the switching signal is generated for the power devices in each sampling time. The implementation of FS-MPC for the PMSM drive is illustrated in Figure 2. The control implementation can be divided into two parts, as follows:

**Table 1.** Voltage vectors of the three-phase VSI.

| Switching State $[S_a, S_b, S_c]$ | Voltage Vectors $[V_\alpha, V_\beta]$ | Vector Number |
|---|---|---|
| [0,0,0] | [0,0] | 0 |
| [1,0,0] | $[2V_{dc}/3, 0]$ | 1 |
| [1,1,0] | $[V_{dc}/3, \sqrt{3}V_{dc}/3]$ | 2 |
| [0,1,0] | $[-V_{dc}/3, \sqrt{3}V_{dc}/3]$ | 3 |
| [0,1,1] | $[-2V_{dc}/3, 0]$ | 4 |
| [0,0,1] | $[-V_{dc}/3, -\sqrt{3}V_{dc}/3]$ | 5 |
| [1,0,1] | $[V_{dc}/3, -\sqrt{3}V_{dc}/3]$ | 6 |
| [1,1,1] | [0,0] | 7 |

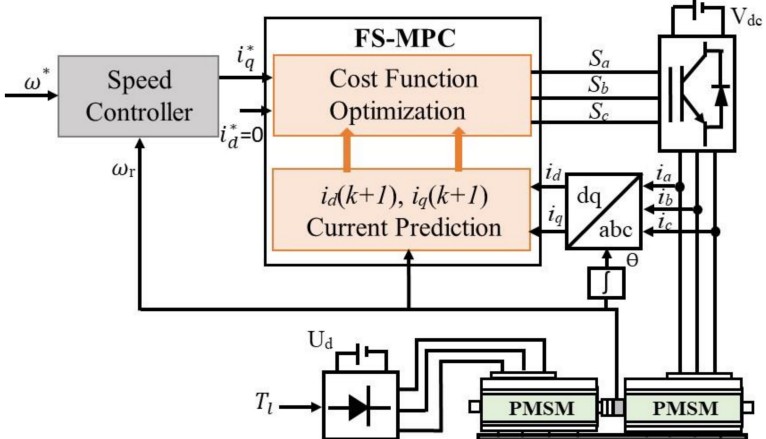

**Figure 2.** Schematic diagram of the finite-set model-predictive control (FS-MPC)-based PMSM drive.

### 2.2.1. Discrete-Time Predictive Model

In this paper, the discrete model of the PMSM is considered as a prediction model for the system implementation. The discrete-time model is used to predict the future behavior of the motor current from the supply voltage and the measured current at the instant of *k*. The discrete-time model can be obtained by an approximation of the load current derivative *di/dt* using the forward Euler discretization method for a sampling time $T_s$, given as:

$$\frac{di}{dt} \approx \frac{i(k+1) - i(k)}{T_s} \tag{8}$$

The discretized motor current on the *dq*-axis is obtained by applying the relation of Equation (8) to Equations (3) and (4), as presented in the following equations:

$$i_d(k+1) = K_1 i_d(k) + K_2 \omega_r(k) i_q(k) + K_3 v_d(k) \tag{9}$$

$$i_q(k+1) = K_1 i_q(k) - K_2 \omega_r(k) i_d(k) + K_3 v_q(k) - K_4 \omega_r(k) \tag{10}$$

where $K_1 = (1 - R_s T_s / L_s)$, $K_2 = T_s$, $K_3 = T_s / L_s$ and $K_4 = \psi_m T_s / L_s$. $i_d(k+1)$ and $i_q(k+1)$ are the motor-predicted currents at the instant of $k+1$. Based on the present sampling current $i(k)$, speed $\omega_e(k)$ and the given stator voltage $v(k)$, the prediction current is evaluated for the next sampling instant $(k+1)$. These equations predict the stator current for each of the voltage vectors of the eight voltage vectors of the inverter.

### 2.2.2. Evaluation of the Cost Function

The cost function can be expressed as the error between the reference current and the predicted current:

$$\begin{aligned} G &= (i_d^* - i_d^j(k+1))^2 + (i_q^* - i_q^j(k+1))^2 \\ j &= 0, 1, .....7 \end{aligned} \tag{11}$$

where $i_d^*$ and $i_q^*$ are the reference stator current on the *d*- and *q*-axes, respectively, $i_d^j(k+1)$ and $i_q^j(k+1)$ are the predicted current corresponding to the inverter voltage vector and *j* represents the number of predicted currents, according to the number of possible switching states. $i_d{}^*$ is considered to be zero in the case of the surface permanent magnet synchronous motor (SPMSM). The first term of the cost function represents the minimization of the reactive power. The second term is used to track the torque-producing current. Finally, the optimized switching states corresponding to the optimized cost function are applied to the inverter circuit.

### 3. XSG-Based Controller Designing

The controller development of the motor drive system consists of two parts: the speed controller and FS-MPC. The implementation of FS-MPC consists of three steps: prediction of the motor current, evaluation of the cost function and generation of the switching signal corresponding to the optimized cost function. The digital designing tool XSG integrated with MATLAB/Simulink is used for designing the controller, which is ultimately used for the FPGA-based system development.

*3.1. Speed Controller*

The controller implantation requires a reference value corresponding to the desired operation. The speed control loop consists of a PI controller to regulate the motor speed and correspondingly generates the reference quadrature axis current $i_q^*$. The implementation of the speed controller in XSG is presented in Figure 3. The XSG-based FPGA implementation requires the discretization of the PI controller. Euler's forward method is used for the discretization of the controller. The discretized equation for the PI control is presented as:

$$
\begin{aligned}
i_q^*(k) &= K_{p\omega}\omega_{er}(k) + x_\omega(k) \\
x_\omega(k) &= x_\omega(k-1) + K_{i\omega}T_s\omega_{er}(k)
\end{aligned}
\tag{12}
$$

where $i_q^*(k)$ and $\omega_{er}(k)$ are the reference $q$-axis stator current and speed error at the instant $k$, and $x_\omega(k)$ is the output of the integral control at the instant $k$. For FS-MPC designed in $dq$ the reference model, the three-phase sensed motor current needs to be converted into its equivalent $dq$-axis current.

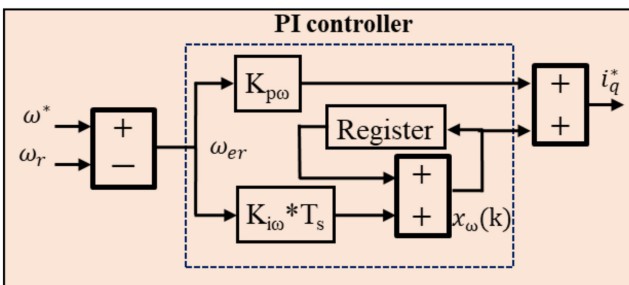

**Figure 3.** Xilinx system generator (XSG)-based design of the speed control loop.

*3.2. Prediction Model*

The discrete-time mathematical equations described in Section 2 are used for the prediction of the motor current. The cost function is computed using an absolute error between the reference current and the predicted current. For the implementation of FS-MPC in the $dq$ frame, the evaluation of the cost function corresponding to the voltage vector $j$ is shown in Figure 4. The cost function $G_j$ can be evaluated for the voltage vector $j$. For a two-level inverter, there are eight voltage vectors and, hence, corresponding to eight voltage vectors, there will be eight cost functions.

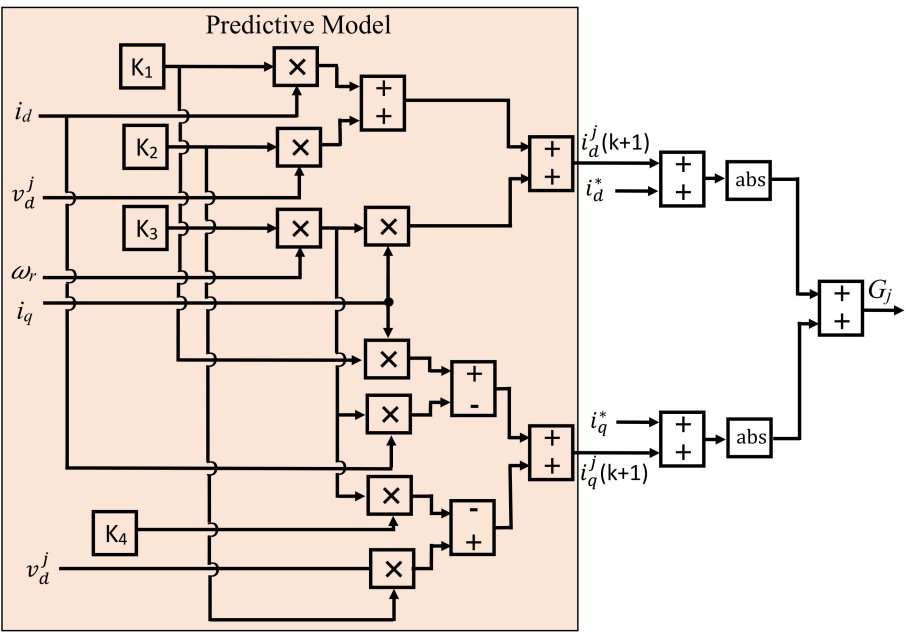

**Figure 4.** XSG-based design of the current prediction and cost function.

### 3.3. Cost Function Optimization and Selection of Switching Vector

The block diagram representing the cost function optimization $G_{opt}$ and the selection of an optimum switching state $S_{opt}$ is shown in Figure 5. In order to compute the optimized cost function, a pipelining method is used. The cost functions are connected to one another to form a pipe-like structure. The two consecutive cost functions in the pipe line structure are compared by a comparator (C), and the minimum among these is selected through a 2:1 multiplexer (M). The output of the comparators (binary digit "0" or "1") is used as select lines ($sel_1$–$sel_7$) for the multiplexers. These combinations of the comparator and MUX (C&M1–C&M7) are used for all the possible cost functions ($G_1$–$G_7$) to select the optimized cost function $G_{opt}$. The switching signals ($S_1$–$S_8$) corresponding to the voltage vectors ($V_1$–$V_8$) are used in a pipe line structure to find the optimized switching signal. The select lines ($sel_1$–$sel_7$) are used for a 2:1 MUX to find the optimized switching signals $S_{opt}$, as shown in Figure 4. Finally, the optimum switching signal is sliced to switching signals $S_a$, $S_b$ and $S_c$. The inverter used for the experimental validation has an inbuilt inversion circuit with a dead time compensation for the PWM signal. Therefore, the switching signals are only generated for the upper-half power devices of the inverter.

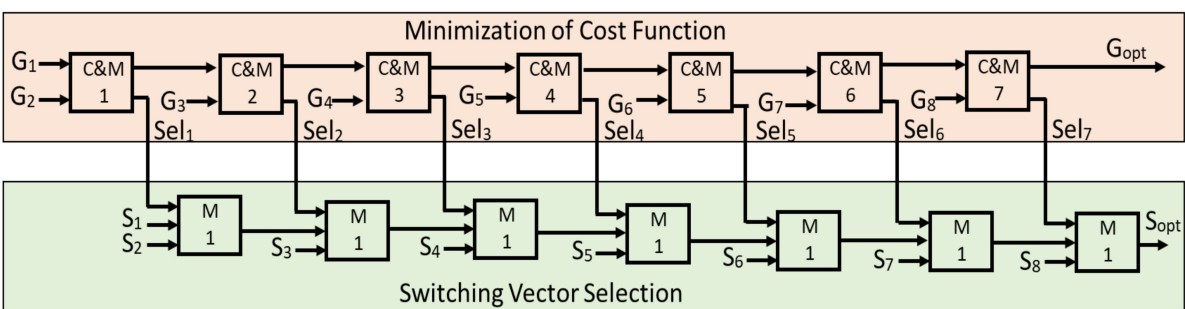

**Figure 5.** Cost function minimization and switching vector selection.

## 4. FPGA-Based System Implementation

The FS-MPC based PMSM drive system in Figure 6 is designed and developed using the digital control environment of the FPGA. The control requires the sensed motor current

and speed for its implementation in the FPGA. The three-phase motor current is sensed through a current sensor and passed through a level shifter. The shifted value of the measured current is fed to the FPGA by using an analog-to-digital converter (ADC). The digitized data from the ADC are synchronized at the rate of the sampling frequency through the ADC interface unit. The encoder is used to sense the motor position and motor angular speed as a pulse, with a resolution of 2048 pulses per revolution. The motor speed and positions can be obtained from this unit by using the position and speed detector. Finally, this parameter is fed to the control unit for the generation of the switching signal. For monitoring and validating, the digital signal is converted to an analog signal by using a digital-to-analog converter (DAC). The control unit, along with the position and speed detector and ADC (DAC) interface unit, is programmed to the FPGA board.

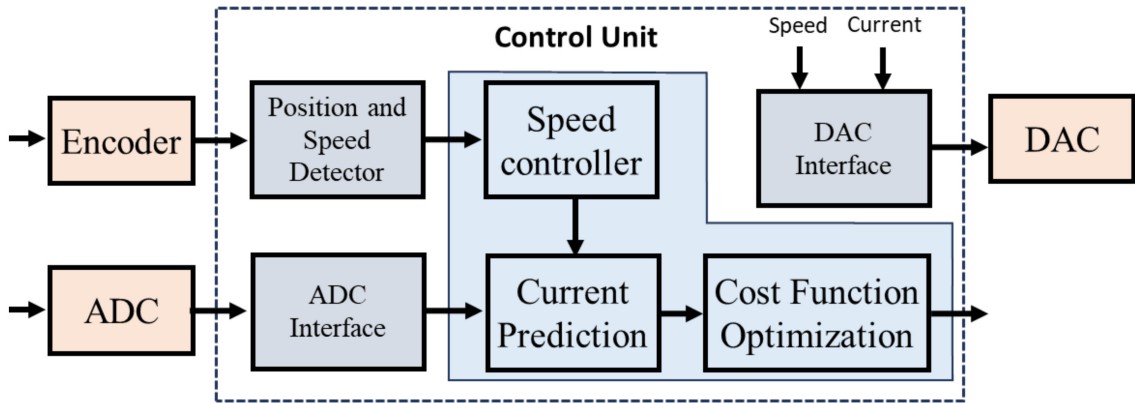

**Figure 6.** FPGA-based system implementation.

For the implementation of the control unit, the output of the speed control unit is responsible for the motor current prediction; therefore, sequential time synchronization is used within a sampling time. The timing diagram used for the FPGA implementation is shown in Figure 7. The start signal has a time of 1 μs, and its width is defined by the sampling time $T_s$. Consequently, the chip select and the done signal were switched to a high state, and the start signal was switched to a low state after the completion of the conversion of the analog signal to the digital signal by the ADC interface unit. The enable signal was generated with an on-time of $T_{on}$ and a delay time of $T_d$. The done signal in Figure 7 as an enable signal enables the speed control loop as well as *abc*-to-*dq* conversion simultaneously to perform parallel computation. The output of the speed control unit ($i_q^*$) and *abc*-to-*dq* conversion ($i_{dq}$) is fed to the FS-MPC unit for the implementation of the control algorithm. Furthermore, the enable1 signal with an on-time $T_{on}$ of 0.5 μs and a delay time $T_d$ of 1.5 μs from the enable signal enables the FS-MPC unit, as shown in Figure 8.

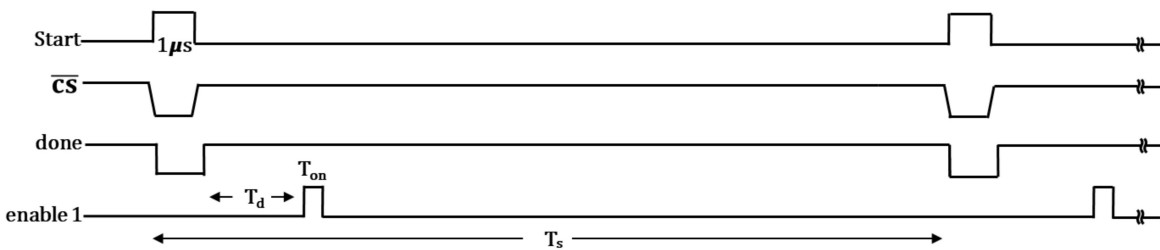

**Figure 7.** Timing diagram of the control loop.

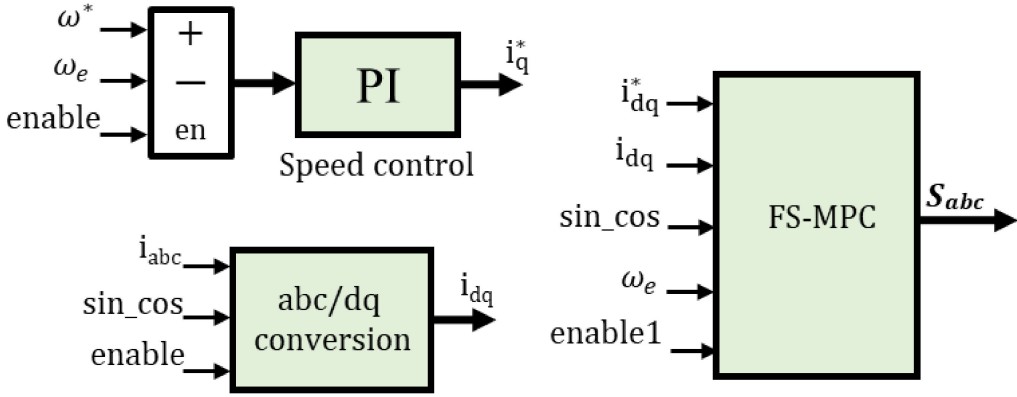

**Figure 8.** Controller implementation in the FPGA.

## 5. Experimental Results and Discussion

An experimental setup, as illustrated in Figure 9, was developed for the real-time implementation and analysis of FS-MPC for a FPGA-based PMSM drive system. The FPGA code was generated automatically through the modeled controller and programmed using dedicated software (Xilinx VIVADO Design Suite) for the real-time operation of FS-MPC. Sampling frequencies of 25 kHz, 50 kHz and 100 kHz were used for the extensive analysis of the impacts of the sampling frequency on the motor performance. The speed controller gain $K_{p\omega}$ and $K_{i\omega}$ values were kept constant for all the cases. However, considering the different sampling frequencies, the sampling time $T_s$ was multiplied by the $K_{i\omega}$ value as: $K_{i\omega} \times T_s$. A step change in the reference speed and motor load condition (load disturbance) was introduced to demonstrate the system performance (current and speed) under the transient operation and also analyzed corresponding to the different sampling frequencies. Moreover, the motor current steady-state error and the current harmonic distortion corresponding to the different sampling times were considered to demonstrate the system performance. The controller was validated by comparing it to a conventional control technique (FOC). The system parameters and the component specifications considered for the development of the experimental setup are explained in Tables 2 and 3. For the FPGA-based system implementation, a clock frequency of 100 MHz was considered.

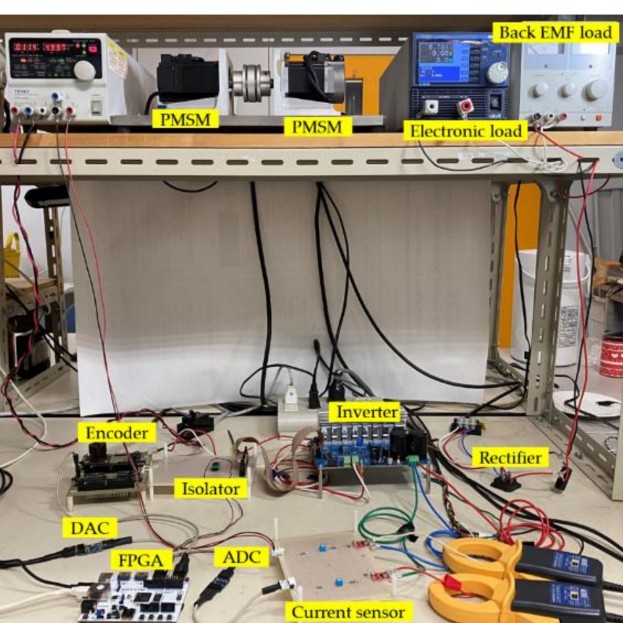

**Figure 9.** Experimental system.

**Table 2.** System parameters.

| Parameters | Values |
|---|---|
| DC voltage $V_{dc}$ (V) | 80 |
| Motor-rated power (W) | 400 |
| Rated torque $T_e$ (Nm) | 1.27 |
| Stator resistance $R_s$ ($\Omega$) | 0.96 |
| Stator inductance $L_s$ (H) | $4.3 \times 10^{-3}$ |
| Permanent magnet flux $\psi_m$ (Wb) | 0.188 |
| Rotor inertia $J$ (Kg-m$^2$) | $5.3 \times 10^{-5}$ |
| Coefficient of viscous friction $B$ (Nm/(rad/s)) | $1.0 \times 10^{-5}$ |
| No. of pole pairs ($p_p$) | 4 |

**Table 3.** Specifications of the system.

| Components | Specifications |
|---|---|
| Three-phase VSI | STEVAL-IHM023V3, 1 kW |
| DC voltage supply | ST5360318 |
| Three-phase rectifier | S15VT60-4000 |
| Electronic load | LSA-165 |
| Back EMF load | PR-18-5A |
| Current sensor | ACS723 |
| Isolator | ADuM3440 |
| ADC | PMOD AD1 |
| DAC | PMOD DA4 |
| FPGA board | ARTY Z7-Xc7z020 |

*5.1. Change in the Reference Speed*

Depending on the applications, the motor may undergo a change in the reference speed. The controller should attain the new reference smoothly with a lower settling time. A step change in the reference speed from 900 rpm to 1200 rpm (low to high speed) and 1200 rpm to 900 rpm (high to low speed) was considered to demonstrate and analyze the performance of the controller for the PMSM drive system.

Figures 10–12 demonstrate the speed regulation of the PMSM for the sampling frequencies of 25 kHz, 50 kHz and 100 kHz, respectively. The response was expanded from 0.4 s to 0.8 s and 2.6 s to 3 s for a close view of the speed dynamic response for a change in the reference speed from low to high and high to low, respectively. The settling time performance of the speed regulation was better for 100 kHz and 50 kHz compared to 25 kHz. With an increase in the sampling frequency, the error between the reference and the predicted current decreases, which ultimately improves the dynamic performance of the speed response. Furthermore, the motor currents $i_d$ and $i_q$ are shown corresponding to the sampling times. Current $i_q$ for the sampling frequency of 100 kHz shows better transient performance. With an increase in the sampling frequency, the controller shows better transient performance. The settling time performance of speed regulation is summarized in Table 4 in terms of the time required to attain the steady state.

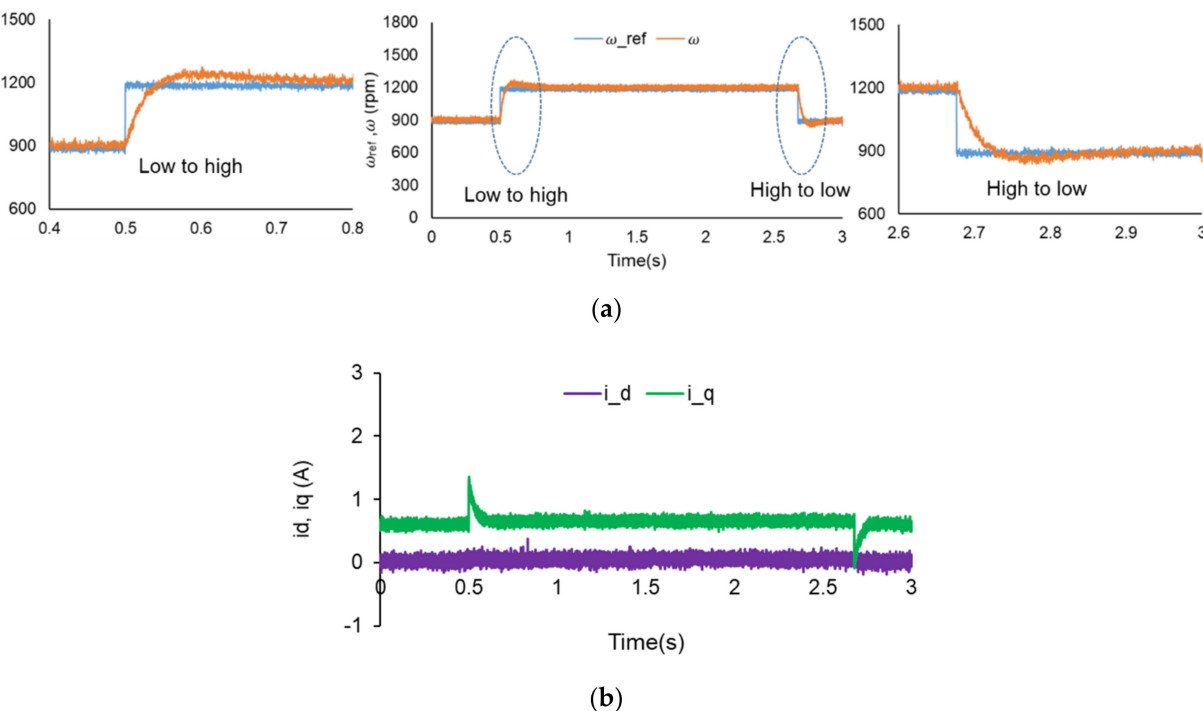

**Figure 10.** (**a**) Motor speed response; (**b**) current response for the sampling frequency of 25 kHz.

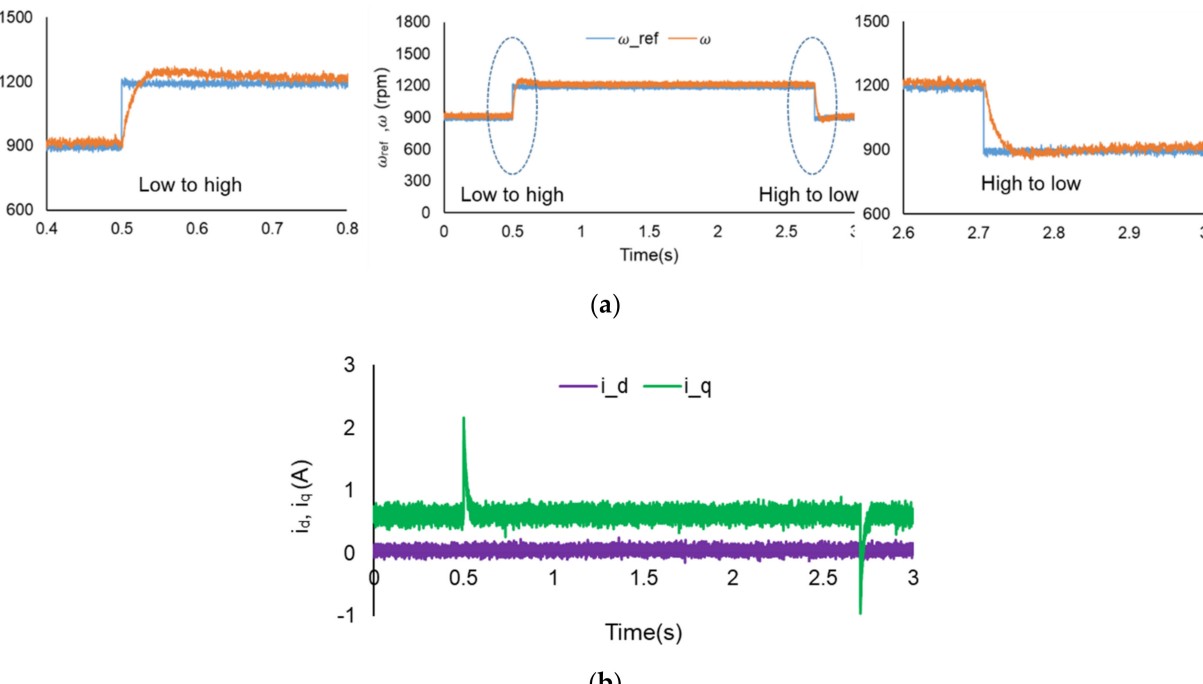

**Figure 11.** (**a**) Motor speed response; (**b**) current response for the sampling frequency of 50 kHz.

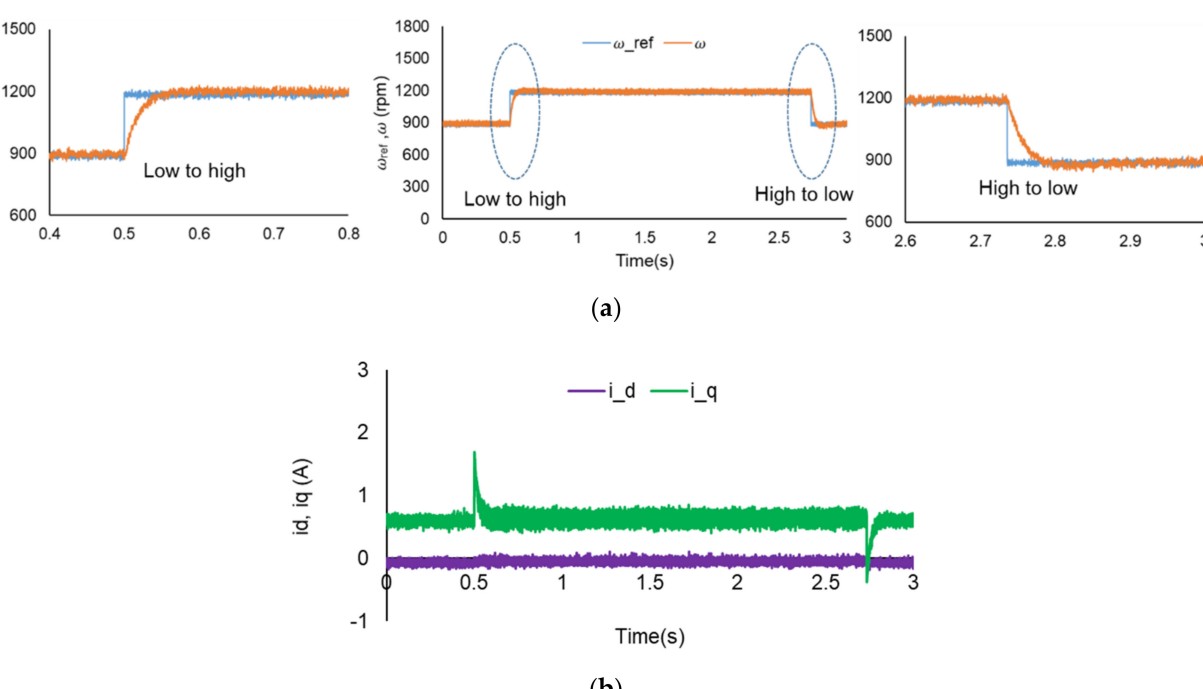

(a)

(b)

**Figure 12.** (**a**) Motor speed response; (**b**) current response for the sampling frequency of 100 kHz.

**Table 4.** Speed response corresponding to the change in the reference speed.

| Sampling Frequency (kHz) | Low–High | High–Low |
|:---:|:---:|:---:|
| 25 | 0.3 s | 0.08 s |
| 50 | 0.2 s | 0.05 s |
| 100 | 0.05 s | 0.025 s |

### 5.2. Change in the Load Condition

The motor system can go under the load disturbance condition as well, and the transient performance of the motor drive system was of concern to attain the desired reference speed smoothly and with a lower settling time. A step change in the electronic load to introduce a load disturbance was employed, which ultimately resulted in a motor current change from 0.5 A to 1 A (low to high) and 1 A to 0.5 A (high to low). The motor speed was kept at 900 rpm for all the operating conditions.

The speed regulation of PMSM under load disturbance in Figures 13–15 is demonstrated for the sampling frequencies of 25 kHz, 50 kHz and 100 kHz. The response is expanded for a close view of the speed dynamic response for a change in the load disturbance from low to high and high to low. The settling time performance of the 100 kHz sampling frequency shows better performance for both high to low load and low to high load. The settling time performance of the speed under load disturbance is summarized in Table 5 considering the time required to attain the steady state.

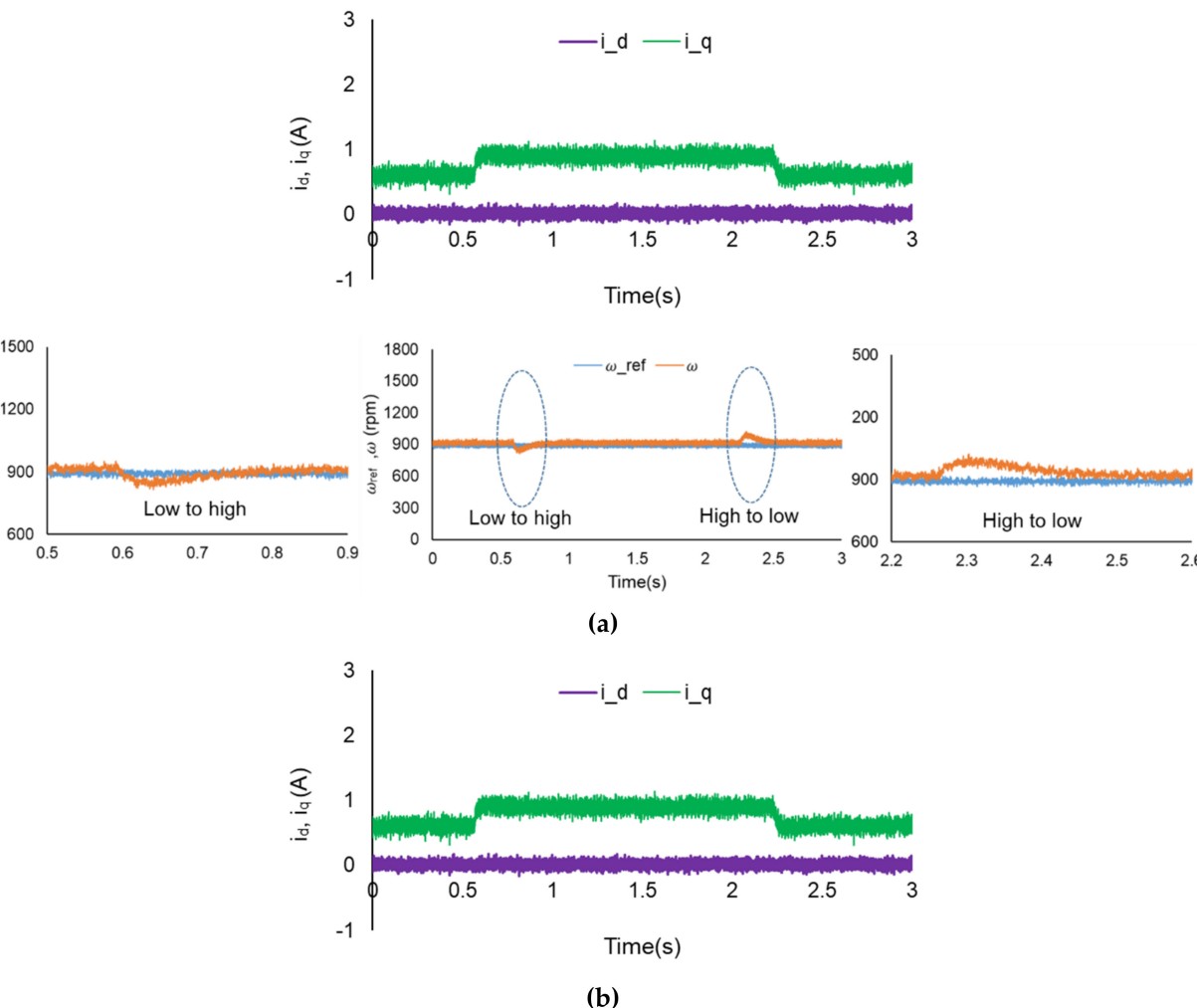

**Figure 13.** (**a**) Motor speed response; (**b**) current response for the sampling frequency of 25 kHz.

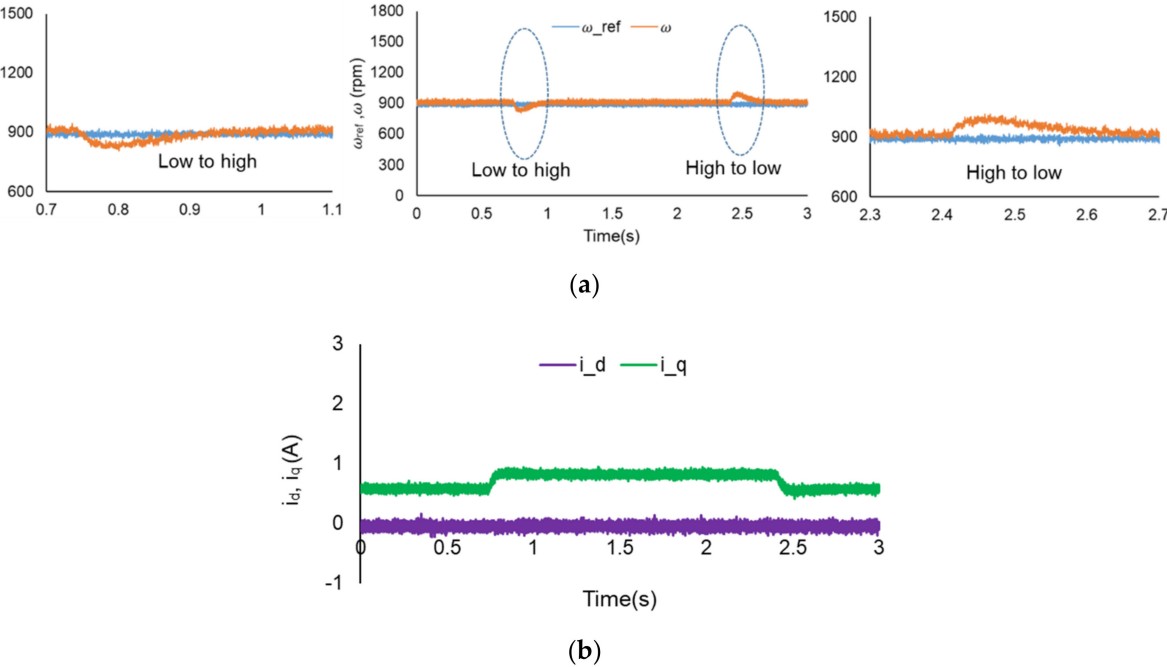

**Figure 14.** (**a**) Motor speed response; (**b**) current response for the sampling frequency of 50 kHz.

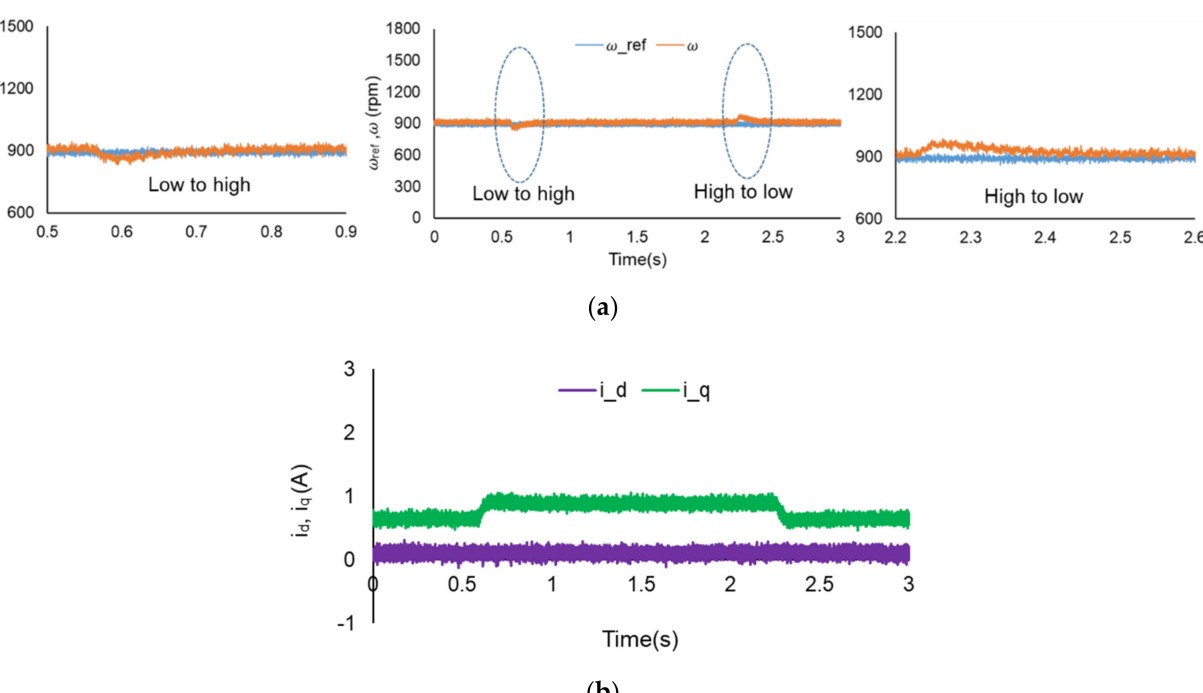

**Figure 15.** (**a**) Motor speed response; (**b**) current response for the sampling frequency of 100 kHz.

**Table 5.** Speed response corresponding to the change in load.

| Sampling Frequency (kHz) | Low–High | High–Low |
| --- | --- | --- |
| 25 | 0.15 s | 0.26 s |
| 50 | 0.18 s | 0.24 s |
| 100 | 0.04 s | 0.17 s |

*5.3. Steady-State Error Analysis*

In order to analyze the accuracy of the controller, the steady-state error (SSE) of the motor current was considered. A comparative analysis of the SSE was performed corresponding to different sampling frequencies through bar graphs. The SSE is calculated by using the below equations [30]:

$$SSE = \frac{\sqrt{\bar{e}_d^2 + \bar{e}_q^2}}{\sqrt{i_d^* + i_q^*}} \times 100 \tag{13}$$

where $\bar{e}_d$ and $\bar{e}_q$ are the mean values of the current error of the *dq*-axis. This error can be calculated as follows:

$$\bar{e}_d = \frac{1}{N} \sum_i \{i_d^*(j) - i_d(j)\} \tag{14}$$

$$\bar{e}_q = \frac{1}{N} \sum_i \left\{i_q^*(j) - i_q(j)\right\} \tag{15}$$

where, *N* is the number of the current samples considered for the calculation of the SSE. The SSE comparison for the sampling frequencies of 25 kHz, 50 kHz and 100 kHz is shown in Figure 16. The motor was working at a speed of 900 rpm. The motor currents of 0.8 ampere were considered for the calculation of the steady-state error of motor current. The SSE with sampling frequency of 100 kHz and current of 0.8 ampere have the lowest steady

state error. The steady-state error decreases with the increase in the sampling frequency and motor current.

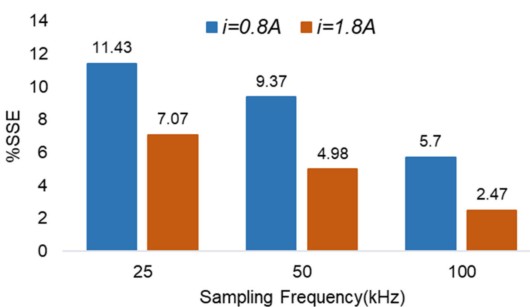

**Figure 16.** Steady-state error (SSE) performance for FS-MPC.

*5.4. System Performance in Terms of THD*

The controller performance (three-phase current harmonics) depends on the switching frequency at which the power devices operate. In the case of FS-MPC, the switching frequency of the power devices is governed by the sampling frequency. Therefore, different sampling frequencies were considered to examine and analyze the controller performance. The maximum switching frequency of FS-MPC is half of the sampling frequency considered for the system implementation. The maximum switching frequency that could be taken for the inverter to use in the experimental system is 50 kHz. Considering this condition, a maximum sampling frequency of 100 kHz can be used for the system implementation. Different sampling frequencies were exercised to investigate the system performance, as shown in Table 6. For all the cases, the motor was operated at a speed of 900 rpm. A current of 0.7 A was considered for the low-load case and 1.2 A for the high load case.

Under higher harmonic conditions, the motor has more iron losses, which ultimately reduce the motor efficiency and service life. Moreover, with an increase in the harmonics, the motor can have noise and vibration. As the switching frequency increases, the harmonics in the current decrease. Therefore, for better performance, the controller should work at a higher switching frequency, meaning a higher sampling frequency in the case of FS-MPC. However, with an increase in the switching frequency, the switching losses increase, which ultimately increases the size of the converter. According to the IEEE standard, a THD current of less than 5% is acceptable. Therefore, by considering both the system performance and the losses, the sampling frequency of 50 kHz can be a better option.

**Table 6.** THD comparison.

| Sampling Frequency (kHz) | % THD for Low Load | %THD for High Load |
|:---:|:---:|:---:|
| 10 | 12.6 | 8.02 |
| 25 | 6.2 | 4.1 |
| 50 | 3.7 | 2.4 |
| 100 | 2.3 | 1.3 |

*5.5. Comparison with FOC*

In order to validate the system behavior of FS-MPC, the dynamic speed response was compared to that of the linear control, FOC. The sampling frequency for both the controllers was kept the same; that is, 100 kHz. The speed controller gain $K_{p\omega}$ and $K_{i\omega}$ were considered to be the same. The values of the controller gain (speed control and current control) are illustrated in Table 7. The controller gain value was calculated from the current controller bandwidth by using the following equation:

$$K_p = \frac{T_i}{2K_{pwm}(1/R_s)T_\Sigma} \tag{16}$$

where $T_i$ is the electromagnetic time constant of the motor, which can be represented as the ratio of the stator inductance and stator resistance ($L_s/R_s = K_p/K_i$); $K_{pwm}$ is the equivalent gain of the PWM inverter; and $T_\Sigma = 1/2f_{sw}$. $f_{sw}$ is the switching frequency considered for the system implementation.

**Table 7.** Controller parameter.

| Parameter | Value |
|---|---|
| $K_{p\omega}$ | 3 |
| $K_{i\omega}$ | 30 |
| $K_{pd}$, $K_{pq}$ | 5 |
| $K_{id}$, $K_{iq}$ | 20 |

The comparative regulatory behavior of the speed control for the change in speed from 900 rpm to 1200 rpm is illustrated in Figure 17a for a sampling frequency of 100 kHz. In the case of FOC, the overshoot for the speed change is higher compared to that of the FS-MPC. Moreover, the settling time is also more compared to that of the FCS-MPC. Similarly, the speed response corresponding to the change in the load disturbance is also illustrated in Figure 17b. The speed response in the case of FOC has a slower response compared to that of the FS-MPC. Moreover, the speed undershoot and overshoot are more than twice as compared to that of FS-MPC. Further, a detailed comparison of the two controllers is presented in Table 8.

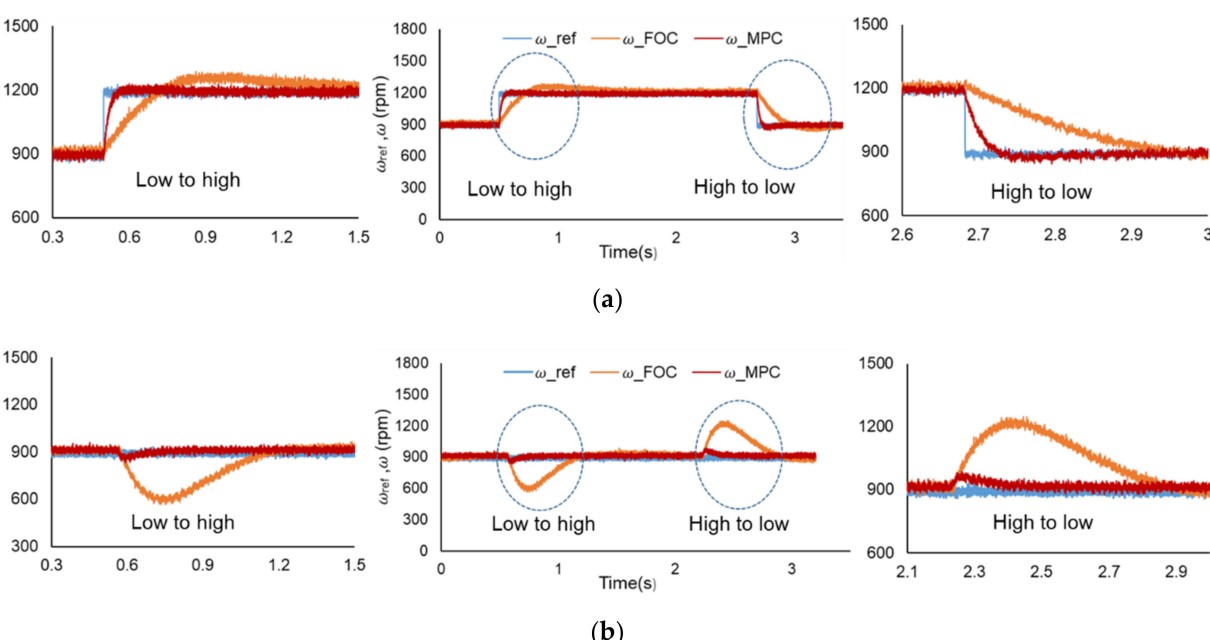

**Figure 17.** Motor speed response for the sampling frequency of 100 kHz. (**a**) Change in the reference speed; (**b**) change in the load disturbance.

**Table 8.** Comparison of controllers.

| | FOC | FCS-MPC |
|---|---|---|
| Speed controller | PI | PI |
| Inner controller | 2 PI | Cost function |
| Tuned parameter number | 6 | 2 |
| Pulse-width modulation | Yes | No |
| Switching frequency | Constant | Variable |
| Computational burden | Low | High |
| Inclusion of system constraints | Difficult | Easy |

## 6. Conclusions

A FS-MPC-based PMSM drive system is presented for the FPGA-based real-time control implementation using the XSG digital simulator integrated with MATLAB/Simulink. In addition, a 100 kHz sampling frequency system for the FS-MPC and its implementation is achieved using FPGA. The sampling frequencies have a significant impact on motor performance under transient operation. The controller demonstrates better performance with a sampling frequency of 100 kHz under a change in the reference speed and load condition. System accuracy and the steady-state error in the motor current were analyzed. The steady-state error decreases with an increase in the sampling frequency. Furthermore, the effect of the sampling frequency on THD was also studied and investigated. The current ripple calculated in the form of %THD is lower, corresponding to the higher sampling frequency. As the sampling frequency increases, the switching frequency increases, which ultimately reduces the current ripples. The sampling frequency has an impact on motor performance. With an increase in the sampling frequency, the controller can achieve a better performance in terms of transient response under a change in the reference speed and motor load condition. In addition, the controller performance is also compared to the FOC PI-PI controller. A faster response for FS-MPC is obtained compared to that of FOC, which ultimately validates the proposed method.

**Author Contributions:** The manuscript preparation, including system design and experiments, was performed by I.M. The idea of the analysis was suggested by R.N.T., and V.K.S. contributed to the manuscript correction. T.H. supervised the writing of the paper and experimental work. All authors have read and agreed to the published version of the manuscript.

**Funding:** This research received no external funding.

**Institutional Review Board Statement:** Not applicable.

**Informed Consent Statement:** Not applicable.

**Data Availability Statement:** Not applicable.

**Conflicts of Interest:** The authors declare no conflict of interest.

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
