# Peer review of "Step-by-Step Development and Implementation of FS-MPC for a FPGA-Based PMSM Drive System"

_electronics, doi:10.3390/electronics10040395_

Round 1

Reviewer 1 Report

Overall, this a commendable work that presents an intuitive method for implementing FS-MPC in a PMSM drive system. Whilst the concept of FS-MPC is not entirely new, but the paper clearly shows a practical approach via the adaptation of FPGA.

However, the authors are invited to address the following comments:

  1. There are a few formatting issues:
    1. Align all equation caption to the right,
    2. Correct typos: line 119 (‘below’), line 223 (can ‘be’ obtained…), line 257 (‘FPGA’), line 254 (change ‘examine corresponding’)
  2. Figure 5 could have been better explained with a flowchart, what is Ud and TL* in Figure 2?,
  3. Line 285: it is not explicitly clear on why the settling time performance is better for 100 kHz & 50kHz. The authors are invited to provide more scientific validation of this claim,
  4. Change the caption for Table 4 and 5 to reflects that fact that the analysis os for settling time,
  5. The analysis presented paints a good picture of the dynamic response (i.e. transient response), but how does the proposed scheme affect the stability of the PMSM. Some form of stability analysis is required to justify the overall reliability of this scheme. Moreover, the presented results seem to have huge oscillations, is this normal for PMSM?

Author Response

Dear respected reviewer,

The authors are grateful and would like to thank you for the comments and suggestions that really helped in modifying the paper. The changes that are done in the manuscript are presented in red color.

Thanks and regards,

Ipsita Mishra

Reviewer 2 Report

References are outdated. Therefore, authors have to improve the state-of-art. . Have you validated your concept by mean of experimental prototype or just verified it by emulation?

Author Response

Dear respected reviewer,

The authors are grateful and would like to thank you for the comments and suggestions that really helped in modifying the paper. The changes that are done in the manuscript are presented in red color.

Thanks and kind regards,

Ipsita Mishra

Reviewer 3 Report

The authors try to present a method for PMSM Drive System through FPGA. Even though the authors present an interest work, it has weaknesses. In particular : 

1. The reference list is not considering the recent publications of Electronics and Energies MDPI journals. Indicative ones that are in the core of the work presented here are : 10.3390/electronics9111762 , 10.3390/electronics9060887, 10.3390/en13215762, 10.3390/en12030443, 10.3390/electronics9091427, 10.3390/en13153975 ,  etc, and there are 147 papers in the fields of motor drive systems and at least 25 are exactly in the field of motor drives for permanent magnet machines.

2. The authors must update their work considering the work presented at least in the above-mentioned papers list, and enriched with additional ones, as theses above are papers of 2019 and 2020 mostly, thus more recent than the ones the authors use in their work.

3. They must compare their work with the work in the above-mentioned papers and to clearly indicate the advantages and disadvantages of their proposal against the existing technologies. This is of paramount importance.

4. The error analysis is missing and is of paramount importance to allow the readers to determine the accuracy of the results and the error sources. This will increase the value of the work presented in this manuscript.

5. The THD issue is critical and affecting the operation of the system as well as the network. The analysis related to this issue and the results presented are very poor. The authors should present the issue and the related measurements in depth to allow the readers to understand how this system responds in the typical rich in harmonics environments that these systems are operating. 

6. In all above issues the comparison with the established techniques is not as extended as it should be and the authors seems to focuses only in strengths and not in the weaknesses of the proposed system. To this they are encouraged to deliver a number of tables presenting the comparison and the strengths and weaknesses of their proposal.

Author Response

(The authors gave the same response as above.)
